# Conceptualizing Task Force Sustainability

## Jennifer Paul Ray

Greater New Orleans Human Trafficking Task Force, Covenant House New Orleans,
New Orleans, LA 70112, USA; jray@covenanthouse.org

**Abstract:** In the anti-human trafficking movement, multi-disciplinary teams have emerged as a best practice for collaborating and coordinating efforts in combating human trafficking. Many multi-disciplinary teams are comprised of federal, state and local partners representing law enforcement, prosecutors and service providers. The concept of sustaining the multi-disciplinary teams is a relatively new area of discussion in the anti-human trafficking movement. This paper explores the Greater New Orleans Human Trafficking Task Force sustainability process as an illustrative example to shed light on the issues that arose during the process for this Task Force, and which may be salient for other Task Forces. This retrospective presentation of the comments and observations made by the Greater New Orleans Human Trafficking Task Force members suggest emerging themes that may help to clarify the concepts other Task Forces should consider and to predict the sustainability outcomes. The members' accounts incorporated in this article are presented as valid points of view for framing conclusions that may be applicable in other contexts and to further the conversation in this understudied area of sustainability. The key focal points related to sustainability explored in this conceptual paper include leadership, funding, collaboration, trust and relationship building, and change is constant.

**Keywords:** human trafficking; multi-disciplinary team; sustainability planning

## 1. Introduction

Evidence of human trafficking in the Greater New Orleans—Metairie Metropolitan area [1], with the city of New Orleans at the center, was nationally recognized in the aftermath of Hurricane Katrina [1]. At the 2006 National Conference on Human Trafficking in New Orleans, Attorney General Alberto R. Gonzalez, announced additional Department of Justice (Department) funding for the establishment of Trafficking Task Forces to identify and assist victims of human trafficking and to apprehend and prosecute those engaged in trafficking offenses [2,3]. The Louisiana Commission on Law Enforcement (LCLE) was one of the award recipients [4]. It received the funding as part of a larger effort by the Department to restore the criminal justice infrastructure in the city of New Orleans (coterminous with Orleans Parish) and to better equip the local law enforcement agencies [2,5]. The funding focused investigations on trafficking activity in the Hurricane [Katrina]-damaged areas of the state, including the Interstate Highway-10 (I-10) corridor in Louisiana, linking the city of New Orleans to Baton Rouge [3,6]. Through the Trafficking Task Force, the Department intended to "put a stop to the exploitation and abuse of laborers", estimated to be thousands of migrant and unskilled workers brought into the city of New Orleans by traffickers to work on the rebuilding efforts [3,6,7]. Authorities believed that these traffickers exploited victims by bringing in laborers, taking documents from them, not paying them and bringing in prostitutes to service the workers in the direct aftermath of Hurricane Katrina and leading into 2006 [6,8].

Under the Department funding period of 2006–2009, the LCLE established the multidisciplinary Louisiana Human Trafficking Task Force (Louisiana Task Force), which included the Louisiana Sheriff's Association, Louisiana State Police, Metro Centers for Community Advocacy (formerly Metropolitan Center for Women and Children), the U.S. Attor-

ney's Office and the Federal Bureau of Investigation [6]. Under the grant, the Louisiana State Police was involved in a number of collaborations with other law enforcement agencies. This included information sharing meetings with "representatives from the United States (U.S.) Attorney's Office, sheriffs from all over Louisiana, the state police, and citizens" as well as providers [6]. Concurrently, the state of Louisiana passed legislation criminalizing human trafficking and directing law enforcement agencies to adopt training programs to enhance the identification and interdictions efforts [8].

In the anti-human trafficking movement, multi-disciplinary teams have emerged as a best practice for collaborating and coordinating efforts in combating human trafficking. Many multi-disciplinary teams are comprised of federal, state and local partners, representing law enforcement, prosecutors and service providers. The concept of sustaining the multi-disciplinary teams is a relatively new area of discussion in the anti-human trafficking movement. Ref. [9] This paper explores the Greater New Orleans Human Trafficking Task Force sustainability process as an illustrative example to shed light on the issues that arose during the process for this Task Force, and which may be salient for other Task Forces. Ref. [10] This retrospective presentation of the comments and observations made by the Greater New Orleans Human Trafficking Task Force members suggest the emerging themes that may help clarify the concepts other Task Forces should consider and predicts the sustainability outcomes. The members' accounts incorporated in this article, as valid points of view for framing conclusions, may be applicable in other contexts and to further the conversation in this understudied area of sustainability. Ref. [11] The concepts formed out of these conversations shape the scope of this paper and provide insights into how the collaboration sustained its leadership, relationships and trust under constant change.

## 2. History of the Task Force

By 2008, many aspects of law enforcement in the city of New Orleans remained in a state of crisis. Almost every aspect of the local criminal justice system was damaged or destroyed by Hurricane Katrina, and the rebuilding was slow [12–14]. Despite this, law enforcement and victim service providers in the Greater New Orleans—Metairie Metropolitan area formed a multi-disciplinary Task Force under the leadership of the U.S. Attorney's Office Eastern District of Louisiana (USAO) [15,16]. The local partners that collaborated under the Task Force included the Jefferson Parish Sheriff's Office, New Orleans Police Department, Covenant House New Orleans, Catholic Charities of New Orleans and New Orleans Family Justice Center [15]. The Federal partner was the Federal Bureau of Investigation (FBI).

During this period, the Jefferson Parish Sheriff's Office expanded its role and joined the statewide anti-human trafficking efforts along the I-10 corridor, led by the LCLE in partnership with the Louisiana's Sherriff's Association [15]. At this time, one of the most identifiable problems regarding human trafficking, especially sex trafficking, relates to the number of hotels and motels within the Greater New Orleans-Metairie area unincorporated jurisdiction [15]. The Jefferson Parish, adjacent to the Orleans Parish, recorded over eighty hotel and motel establishments, which many traffickers preferred for conducting sex trafficking activities, as the nightly rates were cheaper. In response to this activity, the Jefferson Parish Sheriff's Office formed a partnership with the Greater New Orleans Hotel and Lodging Association, which resulted in the Jefferson Parish, Ordinance Chapter 17.5 Lodging Accommodations, which penalized hotel and motel owners for financially benefitting from a sex trafficking venture [17–19]. The adoption of the Ordinance brought awareness of the issue to the city of New Orleans, and subsequently, new law enforcement investigations were designed to identify hotel and motel owners who were financially benefiting from sex trafficking ventures. The investigations led to a criminal prosecution under the USAO and became the first case in the country in which a hotel owner was indicted under this theory [15].

Over the next six years, the awareness about human trafficking grew among state legislatures, who passed laws such as House Bill 49 in 2012 and Senate Bill 88 in 2013,

establishing Louisiana's improvements in criminalizing child sex trafficking, strengthening criminal penalties and adding protection and services for victims [20–23]. Act 564 of the 2014 Legislature was passed, requiring the Louisiana Department of Children and Family Services to compile and publish an annual statistical report on human trafficking in Louisiana [24]. The 2015 Department of Children and Families Annual Report published activities for the 2014 calendar year, reporting the Orleans Parish as the overall lead in the number of trafficking cases for juveniles and adults [25]. Of the 206 confirmed and prospective victims reported in the state, 172 (84%) were sex trafficking victims, 23 (11%) labor trafficking victims, and two were victims of both labor and sexual trafficking [25]. The Orleans Parish reported 96 victims, of which 46 were juveniles, 48 were adults and two were unknown [25] [2].

During the same year, Covenant House New Orleans collaborated with the Modern Slavery Research Project at Loyola University New Orleans to talk to ninety-nine homeless youth about their experiences of labor exploitation [26]. The study produced a report that "focuses on the particular risks associated with homelessness in city of New Orleans and on recommendations provided by the youth for how the community can improve their response to trafficking" [26]. Of the 99 youth interviewed, 14% of the respondents were identified as victims of some form of trafficking, with 11% of the total population having been trafficked for sex and 5% for other forced labor [26]. The study concluded by estimating that approximately 86 residents per year at Covenant House New Orleans were likely to be victims of human trafficking [26].

The data reported by the Louisiana Department of Children and Family Services for the subsequent reports, 2016–2021, delineates the activities in the Orleans Parish, which represents an average of 37% of the total combined juvenile and adult cases in the state [25]. For the same period, the state reported 91% sex trafficking victims, 3% labor trafficking victims, 3% sex and labor trafficking victims and 4% not reported [25] [3].

A 2016 study provided more insight into the prevalence, by estimating a range of between 650 and 1600 trafficked persons in the Greater New Orleans—Metairie Metropolitan area [27]. Highlighting the significance of this estimation by concluding that the Greater New Orleans—Metairie Metropolitan area is likely to have, proportionally, the highest rate of trafficking as the most populous urban area in the state [27,28].

Between 2016 and 2022, the Greater New Orleans Human Trafficking Task Force (Task Force) reports providing client services to over 700 trafficked persons and conducting over 670 investigations, which resulted in over 250 related arrests and the identification of more than 290 confirmed victims [29]. Of the clients served, 69% were victims of sex trafficking, 16% were victims of sex and labor trafficking, 9% were labor trafficking victims and 2% were reported as unknown [30].

The relative increase in labor trafficking cases under the Task Force are partially the result of law enforcement, including the Jefferson Parish Sheriff's Office's collaboration with the Federal law enforcement partners and local ordinance enforcement agencies, conducting a coordinated long-term investigation that implemented new strategies shared through training, technical assistance and cross-agency planning. The investigations into industries such as massage establishments were productive, with one multi-agency coordinated investigation resulting in arrests, the recovery of eight Foreign National victims and the seizure of over 170,000 USD [31]. The lessons learned after Hurricane Katrina led human trafficking investigators to become more engaged in trying to locate any signs of human trafficking conducted through the shelters and to identify any labor trafficking through the numerous amounts of contractors that arrived in the area to assist after Hurricane Ida, which made landfall on the sixteenth anniversary of Hurricane Katrina [32].

The most common countries of Foreign National citizenship of trafficking victims receiving services through the Task Force in the last seven years were Honduras, Nicaragua, Mexico, the Philippines, Guatemala and Saint Lucia. The immigration status of the trafficking victim clients upon entry into the US for all trafficking types was no documentation [30]. The types of exploitation experienced by the clients include prostitution, stripping/exotic

dancing, pornography, construction/landscaping, other, drug trafficking/dealing, domestic servitude, escort service, panhandling, hotel/hospitality services, cleaning services, field labor, restaurant/food services, child care/day care, servile marriage, retail sales and transportation services [30].

## 3. Sustainability Process

Commencing in 2015, the multi-disciplinary Task Force, led by the USAO, was awarded funding under the Department's Office for Victims of Crimes, FY 2015 Enhanced Collaborative Model to Combat Human Trafficking (Enhanced Collaborative Model) solicitation until September 2018 [32]. The Task Force restructured to a co-leadership model as the Greater New Orleans Human Trafficking Task Force with the Jefferson Parish Sheriff's Office as the lead law enforcement agency and Covenant House New Orleans as the lead victim service provider [15]. The co-leadership formed the Task Force Core Team (Core Team) with Homeland Security Investigations (HSI), the USAO and the FBI. Case management funding was awarded to three victim service provider sub-recipients, and a network of twenty service providers was established under a memorandum of understanding. The collaboration and coordination case management efforts led to the identification of victims and survivors referred to the comprehensive services network through a "No Wrong Door" policy. The strategic partners that participated in the memorandum of understanding include local, state, and Federal law enforcement; judicial district and Federal prosecutors; local and regional direct service and housing providers; health system providers; health department; regional child advocacy center(s); culturally specific organizations; legal service providers; workforce services, as well as mentoring and youth advocacy support programs. The Task Force continued to function under the Department's Enhanced Collaborative Model when it received its second round of funding from the Office for Victim of Crime in October 2018 and continued until January 2022 [33].

The success under the Enhanced Collaborative Model informed the Core Team leadership decision to sanction evaluations, group conversations, qualitative interviews and a self-assessment survey to be conducted for developing a Task Force sustainability plan. The purpose of the sustainability plan was to formalize a path for achieving the long-term goals and to document the strategies that support the Task Force activities and partnerships.

The sustainability process, conducted during COVID-19 over a twenty-two month period (August 2020–May 2022), commenced under the direction of the Task Force evaluator consultant, who guided Task Force discussions around sustainability by conducting qualitative interviews with members of the Core Team leadership, law enforcement, direct service providers and other members of the Task Force network [34]. The Task Force evaluator reported: "There is much to suggest that the Task Force has a high level of sustainability" [34].

As shared through the qualitative interviews, members reported that strong collaboration was the most frequently cited strength of the Task Force [34]. An increase in trust through networking and protocol implementation between service providers and law enforcement, as well as a growing reliability on one another to diversify the services offered to clients, are indicators that the Task Force exhibits the Department outcome criteria for Task Force sustainability [34].

The Task Force evaluator consultant recommended that the Task Force devise and monitor a strategic plan that focuses on sustainability to ensure the continuation of the operations for the Task Force [34]. In response, the Task Force consulted a number of models endorsed by the Department to qualify the scope of sustainability and planning process under the Department's enhanced collaborative model approach and to establish the best practices for conducting its sustainability planning process [35–37]. For instance, the Task Force utilized the "Sustainably Toolkit as a model for the Sustainability Plan: Enhancing Community Reponses to the Opioid/America's Addiction Crisis: Serving our Youngest Crime Victim" (Toolkit) published by JBS International [35]. The Task Force implemented the Toolkit's sustainability planning process, including the assembling of

a planning team, conducting a sustainability assessment, reviewing and discussing the findings, summarizing and prioritizing the findings and developing an action plan through objectives and goals, monitored through bi-annual updating of the plan and reporting to the Department [35].

The Task Force also participated in the Department's self-assessment survey for multi-disciplinary teams to measure their success and to assess its performance through the stages of group development [36]. The results of the Task Force's self-assessment revealed that the multi-disciplinary team was poised for sustainable growth based on the Department stages of group development [38].

Feeling encouraged, the Core Team contracted a second consultant, who challenged the Core Team and members to ask what would happen if the Task Force ceased to exist. By shifting the questions towards the loss of the Task Force through open-ended questions, the sustainability of the Task Force emerged as a personal question of how one would be impacted in their work if the Task Force no longer existed. The analysis of the responses also revealed that relationships are important to the Task Force. In addition, the different levels of engagement at different times by the Task Force members was also learned. This shifted the expectations for involvement and led to the understanding that the Task Force's strength is measured across its whole, with its varying degrees of membership participation [39].

One weakness identified through this process was the potential loss of funding for the Coordinator position because the Task Force activities and administration were conducted by a full-time paid position. "The coordinator is the sustainer. If there is not one person whose job it is to arrange all of the moving parts, then the Task Force is going to fall apart. That person needs to be there to keep it going. We all have so much on our plate that we can't add anything to it" [34].

Assessments of the key issues were also incorporated. In many instances, the assessments were conducted by partners who shared documentation that could be analyzed as part of the sustainability plan process. The partners who conducted these assessments with the Task Force members included UNITY, of Greater New Orleans [40], the Freedom Network [41] and Polaris [42]. The state of Louisiana study on its challenges and gaps in addressing human trafficking was applicable to the assessment because the reported findings were due, in part, to the interviews conducted with Task Force members [23]. An environmental scan on publications related to anti-human trafficking activities in the Greater New Orleans—Metairie Metropolitan area was conducted for historical context. National guidance was also provided through the Human Trafficking Capacity Building Center [43].

The shared feedback, assessments and subsequent analyzes along with the findings from the ongoing conversations and efforts, were synthesized through a strength, weaknesses, opportunities and threats (SWOT) analysis, and informed the development of the priority areas [35]. The findings were used to develop the sustainability plans priority areas, goals and action items. In addition, to ensure the sustainability plan was reflective of the Task Force membership, a draft was presented at a public meeting and circulated to the Task Force membership and interested community members for comment [44]. The membership comments about the Task Force's effectiveness and function were compiled and incorporated as valid points of view into the finalized draft submitted to the Office of Victims of Crime for final review and approval [44].

## 4. Self-Assessment

The history of the Task Force, starting with its origins up to is current leadership, exhibits how the response to human trafficking in the Greater New Orleans—Metairie Metropolitan area and the state of Louisiana is connected through a dynamic flow of members and community champions, who have emerged at critical moments. This history reinforces to the Task Force that change is constant. Observations made by Campbell and Fehler-Cabral [11] reinforce this predictor of "change is constant" as a characteristic of multi-disciplinary collaboration (para.25). Further emphasizing this point, Campbell

and Fehler-Cabral suggests [11] that implementing an inter-agency multi-disciplinary response is a" long-term endeavor", and there is an expectation that there will be staff turnover, shifts in leadership and agency priorities, as well as uneven resource capacities among the participating law enforcement agencies and nonprofit organizations over the duration of the initiative (para 3). Compounding this is the impact of COVID-19, which has contributed to an unexpected staffing attrition caused by local workforce shortages and over-extended staff and personnel. In the case of the Task Force, the law enforcement and victim service provider leadership that started the Task Force seven years ago have moved on. The agency relationships established by the original leadership have maintained through memorandums of understanding, but with law enforcement turnover and staff shortages, it is difficult to sustain these relationships without the connecting element of the Task Force (anonymous survey response). As a member observed, "the victim service provider relationship to law enforcement, such as the FBI, would likely cease or be greatly reduced without the Task Force. There won't be as many opportunities to have multiple agencies together, in person, to discuss trafficking".

During COVID-19, this became harder to achieve due to the difficulty of not having everyone present for meetings, which limited face-to-face interaction and created a loss of connections, resulting in a shift in the interpersonal relationships within the Task Force (anonymous survey response). This was challenging to achieve through the virtual platforms because many law enforcement agencies did not have the resources to provide personnel with the equipment and access to virtual platforms. This changed the nature of the interactions at meetings, which experienced limited attendance, contributing to the loss of direct lines of communication with key partners. The value of direct interaction is expressed by a Task Force member, "open communication is the purpose for coming together to share ideas, bring expertise to the table; and, the collaboration helps put the pieces together during operations and investigations".

The Task Force has been successful at sustaining the ebb and flows caused by funding, and the lack of funding, throughout its history. Its evolution has been situated in the anti-human trafficking movement, which tends to focus on the Department's law enforcement and criminal justice approaches, which is commonly referred to as a top down movement, compared to the domestic violence and sexual assault movements, which started at the grass root level [45]. The goal of the multidisciplinary approach is to utilize the information and resources across law enforcement agencies to activate investigations and prosecutions and to operationalize the trauma-informed, victim-centered response by incorporating victim service providers into the investigative strategy that refers recovered victims to stabilization and safety [11].

Federal law enforcement investment in the multidisciplinary team approach in its respective agencies has built capacity and staff in national, regional and field offices [46–48], such as the FBI Child Exploitation and Human Trafficking Task Forces that operate within nearly every FBI field office [49]. The interface between the internal multidisciplinary team and the collaboration across agency jurisdictions through inter-agency multidisciplinary teams is integrated into the Federal strategies. As the FBI states on its public website, part of its national strategy is to effectively "investigate human trafficking through a collaborative, multi-agency approach with our federal, state, local, and tribal partners" [49]. This support from the national infrastructure has had a direct impact on the local Task Force, which bolsters the local field office leadership's capacity to dedicate the staff and align their priorities and goals to support the local, community-coordinated responses [50–52].

At the local level, a transition emerged with its legacy nonprofit organizations. During COVID-19, and through the subsequent years, the local nonprofit leadership who took the position in the era before and after Hurricane Katrina began to retire. The nonprofit boards sought out new leadership to address the long-term vulnerabilities and risks affecting the community because of Katrina, and at the direction of exiting executive directors, invested in emerging local community leaders who had direct experience in the aftermath of Hurricane Katrina. The emerging leadership recognizes that:

These approaches take time to really integrate into systems and agencies. It would require training and policy change, not simply written agreements. It would necessitate a values clarification and collective reflection on the nature of trafficking as part of broader oppressions [53].

The differing approaches between the law enforcement and victim service providers emerge in the context of addressing the broader oppressions and material hardships in New Orleans. As law enforcement focuses on the recovery of victims, victim service providers lean into the core issues that cause human trafficking:

Because human trafficking often persists out of material hardship in New Orleans, I would like to see the funded Task Force use some funds to address those hardships. Whether that is for emergency housing funds, relocation expenses, or other direct assistance, I think some shared funds would bring the agencies together to really consider the barrier that material needs present for victims trying to leave [53].

Victim service providers continue to advocate to policy-makers and decision-makers for broader changes in the region (i.e., raise minimum wage, receive set-aside HANO vouchers, etc.) as being helpful in addressing some of the material conditions of the New Orleans community (anonymous survey response). This sentiment suggests that the focus of the Task Force is its sustainable integration into the community systems. As one member stated: "ultimately, we should clarify the 'task' of the 'Task Force' and add value to the community as an entity, rather than just be a place of coordination between agencies".

Victim service providers affirm that revisiting the interagency agreements and a clarification of the values across the Task Force would generate internal policy changes within the organizations based on agreed shared values, and this will increase the likelihood of sustained development (anonymous survey response). Law enforcement frames this as an "opportunity to revisit and talk through roles and responsibilities through frank conversations to help focus the Task Force and understand mutual goals of partners", as one member explained.

The common ground is provided by the Department's enhanced collaborative model for a multi-disciplinary team, which is the implementation of the program through a trauma-informed, victim-centered approach. The trauma-informed, victim-centered approach is the core principle that defines and informs the Task Force partners on how to implement the model and achieve its objectives through deliverables and performance measures. The self-assessment revealed that both the law enforcement and victim service providers embraced the victim-centered, trauma-informed approach as a mutual foundational value. As one member observed, "the Task Force provides ongoing training, including law enforcement, and if law enforcement wasn't trained or connected to victims service providers through the Task Force, the victims would come to the victim service providers more traumatized".

The Task Force members' responses to the self-assessment agree that the overall anti-human trafficking work of law enforcement would not change if the Task Force ceased to exist because human trafficking, including the dismantling of criminal organizations, are part of the agencies' ongoing work and mission. Conversely, the members also agreed that the collaboration among law enforcement under the Task Force is important and that there would be a loss in certain law enforcement areas when working cases that benefit from coordinated efforts, such as criminal networks, with a nexus between local, state and Federal jurisdictions (anonymous survey response). Emphasizing the successful integration of the Task Force into the daily activities of the partner agencies, one member stated: "professionally all of the collaboration and open communication would change, which is the purpose for coming together to share ideas, bring expertise to the table, and the collaboration, that helps put the pieces together during operations and investigations". The loss of the Task Force would mean law enforcement would not have the direct line of communication with key partners and would lose the avenues for developing new contacts and communicating with victim service providers (anonymous survey response).

The dynamic of the different levels of funding among the Federal agencies and local partners, particularly the local nonprofits, emerged as a significant discussion point during

the sustainability process. Both sectors acknowledged that the Federal agencies had the sustainable capacity to participate in the Task Force, while the loss of funding for nonprofit partners would directly impact the nonprofits' capacity to provide services to victims and survivors. The loss for law enforcement would be professional, while the loss for victim service providers would be infrastructural. This is because the federally funded Task Force closes the gaps in services through the partner nonprofit victim service providers, which maintains a network for referrals that supports services and provides access to the appropriate care that victims and survivors of trafficking require. Here, the successful integration of the partner agencies through the Task Force has created an unintended dependency on the Federal funding, as one member observed:

Law enforcement isn't relying on the funding, while the nonprofits, especially the lead, is tasked with securing sustainable funding for the network of direct services. There is a need to bring other nonprofit providers through a collaborative commitment for the purpose of raising funds to keep the system [53].

The conversations brought about the realization that the potential loss of funding for the nonprofit partners posed limitations to the Federal agencies as well, which relied on the service providers' network of support and services for victims and survivors of trafficking to bolster their trauma-informed, victim-centered approach. This further expresses the value of the integration achieved by the Task Force as it identifies its mutual reliance on partners, as expressed by a member:

Federal collaboration with local service providers is important, and the current network of referrals may be impacted if the Task Force no longer exists. The indirect outcome is that this could limit the Federal agencies' ability to make referrals when assisting victims/survivors potentially re-traumatizing victims/survivors. A loss of funding would also impact the service provider's capacity to assist victims/survivors [53].

This tension disproportionately effects the lead victim service provider due to its central position in the flow of clients and key resources and its administrative function for the Task Force in the local human trafficking response (anonymous survey response). Under its leadership, those who serve in the anti-trafficking field have built a community network of providers, services and resources available to trafficking victims and survivors, which has added to the quality of care received by clients. The leadership has proven effective at creating access with victim service providers through training and outreach, which has brought the awareness and expansion of services, such as the hiring of staff with expertise in the mental health field. The expanded services give clients access to multiple service providers, with many clients seeing two or more of the Task Force member organizations when needed.

During the sustainability process, a call to other local providers was made to collectively identify and apply for the funds needed to keep the system alive, including the domestic violence and sexual assault nonprofit providers that have provided support services to human trafficking victims and survivors under the Task Force. This is significant because the loss of funds for some partner service providers drastically changed the way the organizations were able to serve human trafficking victims. It was no longer sustainable for some to continue providing services to this population if there was no funding (anonymous survey response). Amplifying these concerns, as one member explained, is the understanding that human trafficking is a difficult crime to investigate because it is not a victimless crime and thus requires significant resources to support the investigations and the specialized services needed for victims and survivors.

Concerns about the mission and the loss in traditional funding from other Federal agencies weakened the collaborative philanthropic strategies as most organizations were struggling to reinforce the core programs and as a result of unplanned staffing losses. Historical silos between the domestic violence, sexual assault and anti-human trafficking movements resurfaced as a result of the increase in client violence locally for domestic violence and sexual assault victims during COVID-19 and the perception that the Task

Force's efforts overshadowed the long-standing services established through grass roots efforts [12,45].

In light of these barriers, the members agreed that the Task Force is the backbone entity that receives the funding to support its administrative activities through the lead organizations that take on the Task Force's responsibilities in the community [54]. The responses also revealed that the Task Force is not a true governance structure because the burden of administration falls on the two lead organizations. The interactions within the Task Force illustrated that its network functions through a structure qualified by Provan and Kenis [55] as "operational links that are often dyad" and based in a referral network, the sharing of information, the participation in joint programming and case management and the collaborative and coordinated investigative activities and prosecutorial actions (para 4).

The Task Force members assessing the Task Force's common agenda were able to articulate its effectiveness [55]. This effort resulted in the re-establishment of its boundaries for the Task Force's network, including the affirmation of key roles and discussion about the foundational elements that framed the members' shared values. The affirmation of the leadership by the two lead organizations was upheld because the Task Force's purpose closely aligned to their respective organizations' mission statements. This contributes to its success in effectively closing the gaps in services and ongoing efforts to provide clients access to the appropriate care. The consensus among the members agreed that the Task Force needed to remain centralized under the lead organizations for collecting and reporting data and for enabling the coordination of services through the maintaining of the service provider network. As noted by the members, the Task Force is instrumental in disseminating information and in identifying funding opportunities for the participating agencies in order to build capacity and sustain community engagement.

## 5. Filling the Gaps

The Greater New Orleans Human Trafficking Task Force Sustainability Plan 2022 (Plan) establishes the priorities, goals, objectives and strategies to fill in the gaps identified through the self-assessment (The download link of The Greater New Orleans Human Trafficking Task Force Sustainability Plan 2022 in Supplementary Materials). The Plan's priority areas are collaboration and partnership; coordinated care and investigations; public engagement and advocacy; and a sustainable Task Force. The crosscutting issue throughout the Plan is that relationships are important to the Task Force. In each priority area, strategies are established for building and sustaining interpersonal relationships within the Task Force because, as one member observed, it "Is not due to any individual person but rather a system that is not set up to put those values first".

In response to this, the Task Force established a course for open-communication and thriving relationships to build capacity around cases and investigations and to help address issues and explore solutions [55]. A strategy for achieving this is by conducting multi-sector collaboration exercises and activities for the Core Team, members and community at large [52]. Promoting the collective impact and the role of the Task Force in the Greater New Orleans-Metairie Metropolitan area affirms the collaborative value that emerged as a reoccurring core principle that has sustained the Task Force over the last seven years. The strategic advancement of the Task Force's trauma-informed, victim-centered approach by promoting the Task Force expertise, assets and best practices through outreach and speaking engagements by members reinforces the agency's shared values and recognizes the individual achievements that have contributed to the Task Force's success [44]. Increasing the data availability and accessibility by reviewing the current data collection and data sharing protocols promotes research into local trafficking prevalence and discussions around improving service provisions, access to supportive services and successful investigative strategies [44]. Sharing the data trends and law enforcement roles through coordinated outreach to encourage the community reporting of tips and other suspected activities advances law enforcement's long-term investigations [44]. As a member explains,

ongoing public outreach in the community helps by providing tips to law enforcement because their current information flow is just scratching the surface [44]. Strategies for developing and implementing standardized and coordinated case efforts across disciplines, through the sharing of best practices and lessons learned, also advances law enforcement's long-term investigations [44].

The involvement of victim service providers in ongoing cases will increase through the development and implementation of a coordinated adult human trafficking multidisciplinary team for case review and care coordination that strengthens the comprehensive care model's support of human trafficking victims' and survivors' specialized immediate and long-term needs [44]. Strategies for integrating law enforcement and the district attorney's office, in coordination with the locally established sexual assault response teams and domestic violence multidisciplinary teams, will expand the case reviews. [44].

The Task Force client demographics reveal that the local material hardships particularly impact African-American females, the most served clients through the Task Force, who find themselves coerced into sex trafficking but identify as domestic violence victims [30]. The Task Force data also report an awareness of vulnerabilities within the Foreign National population, representing 13% of the clients served over the past seven years [30]. The numbers shared by the Task Force partners, such as the Louisiana Office for Refugees, support the Task Force reporting and reveal that more Foreign Nationals are seeking support and services for violent crimes, such as human trafficking, in the state, impacting the Greater New Orleans-Metairie Metropolitan area with its relatively large Foreign National population [56]. Hondurans make up the majority of the Central American population in New Orleans and constitute the largest Honduran community in the United States; Mexicans, Nicaraguans, Guatemalans, Salvadorans, and Cubans are other significant groups that make up the Latino/a community [56,57].

Covenant House New Orleans applied for a third round of funding under the Department's Office for Victims of Crimes Enhanced Collaborative Model Task Force to Combat Human Trafficking in 2022 as one of the goals established under the Plan. During this effort, the lead law enforcement agency, the Jefferson Parish Sheriff's Office, experienced the retirement of the long-standing Vice Unit Commander who had led many of the Task Force local law enforcement operations. This led to an unplanned dismantling of the Sheriff's Office human trafficking team. As a result of the established Task Force's long-standing goals to establish a formal partnership with the city of New Orleans law enforcement and District Attorney's Office, the lead victim service provider and Core Team developed a joint application with the Orleans Parish District Attorney's Office. The awarding of the new grant, which commenced on 1 October 2022 [58], reinforced the Task Force's history, which demonstrates that its sustainability is due, in part, to the connected dynamic flow of members and community champions who have emerged at critical moments.

## 6. Discussion

The approach to sustainability planning presented here reflects the process undertaken by anti-trafficking professionals in the Greater New Orleans-Metairie Metropolitan area. The issues they engaged in during this process and the focal concepts that emerged have the potential for guiding conversations amongst multi-disciplinary teams in other regions. The focal concepts revealed the Task Force's values. The values expressed by the members during the sustainability planning process include relationship building, collaboration, information sharing, open communication, facilitated resource sharing, attendance at regularly scheduled meetings, trauma-informed and victim-centered approaches and coordination through the multidisciplinary approach. The underlying foundation to these values is the commitment by its partner organizations and individual members to sustain the Task Force.

This long-term commitment is possible due to the increased trust among the service providers and law enforcement through the Task Force's ongoing networking, mediations and protocol development and implementation [38]. As noted by one Task Force member,

trust is a direct result of strengthening relationships and it is difficult to formally document, but it is an important goal that has a significant impact on the Task Force and the maintaining of its strength, such as building capacity around cases.

Maintaining this level of trust in the Task Force between the law enforcement, prosecutors and service providers, is difficult. Individuals are more comfortable seeking advice and sharing information within their respective fields. To address this effectively, the members supported the goals and objectives established in the Plan that included strategies for building trust and the agreement that revisiting the roles and responsibilities of the multi-disciplinary team members through the implementation of the Plan would help focus the Task Force's activities and understand the mutual goals of its collaborative partners [52,55]. Establishing opportunities for open communication across sectors through in-person meetings and conversations is also a solution because it sets the stage for bringing expertise to the table and helps build trust in the collaborative efforts through the coordinated work of putting the pieces together during investigations while providing comprehensive services to clients [52,59].

This paper's focal points, such as trust, establish the key findings from the Task Force's sustainability planning process. The implications of the focal points are that leadership, funding, collaboration, trust and relationship-building in a landscape of constant change are the tenants needed for a multi-disciplinary team's sustainability. These elements are crosscutting and understood as they interrelate in the multi-disciplinary team's dynamics. The implication of this is that sustainability requires the ongoing and intentional investment in these elements in order for the inter-agency collaboration to be sustained. The methods presented here reflect the current position of the Greater New Orleans- Metairie Metropolitan area professionals in the anti-trafficking field and have the potential for guiding conversations amongst multi-disciplinary teams in other regions.

A review of similar studies in the sexual assault field suggests that these key findings reflect the characteristics identified for multi-disciplinary teams in both fields [11,59,60]. One commonality is the dynamic between law enforcement, prosecutors and victim service providers. In the case of the Greater New Orleans-Metairie Metropolitan area, this holds true. Many of the law enforcement, prosecutor and victim service providers who participate in the Task Force are also members of the sexual assault response team and the domestic violence multi-disciplinary team. A more in-depth study, that was too broad to undertake in this study, is research on how the multi-disciplinary team characteristics carry over from sexual assault to domestic violence and to human trafficking responses in local communities.

The ability to fund the Task Force outside of the Department grant program has eluded its partners. The discussion section reviews the funding dynamics and presents the success and challenges faced by the Task Force's lead nonprofit. For instance, competing for local funds that are historically awarded to Task Force partner organizations that provide domestic violence and sexual assault support and services. Research of private foundations and philanthropic collaborations are promising for supporting collaborative efforts across domestic violence, sexual assault and human trafficking, and require further research [61,62].

## 7. Conclusions

The Task Force has been the driving force behind the collaborative response to human trafficking in the Greater New Orleans—Metairie Metropolitan area. If the Task Force ceased to exist, there would be a gap in the leadership, historical knowledge, communication, quality of programming and services and collaborative networks [55]. The Task Force's decision to develop the Plan through self-assessment provided an opportunity to revisit its history, shared values, relationships, roles and its collective impact on the community [55]. The key findings defined the Task Force's priority areas for framing the Plan, which are collaboration and partnership; coordinated care and investigations; public engagement and advocacy; and a sustainable Task Force [55]. The priority areas' actionable goals,

objectives and strategies are intended as a map for working through the challenges and gaps identified by the Task Force. Noting that not every participating member will have the same scale of impact, but by virtue of involvement by the members at varying degrees and at different times, all members are contributing in a meaningful way to the sustainable anti-human trafficking response in the Greater New Orleans—Metairie Metropolitan area for the purpose of combating human trafficking, providing survivor services and to support the well-being of our community's survivors [55].

**Supplementary Materials:** The Greater New Orleans Human Trafficking Task Force Sustainability Plan 2022 can be downloaded at http://www.nolatrafficking.org/sustainability-plan (accessed on 2 February 2023).

**Funding:** This manuscript was produced by Jennifer Paul Ray, Coordinator, Greater New Orleans Human Trafficking Task Force, under 2018 VT BX K087 and 15POVC-22-GK-03656-HT, awarded by the Office for Victims of Crime, Office of Justice Programs, U.S. Department of Justice. The opinions, findings, and conclusions or recommendations expressed in this manuscript are those of the contributors and do not necessarily represent the official position or policies of the U.S. Department of Justice.

**Institutional Review Board Statement:** Not applicable.

**Informed Consent Statement:** Members were provided the final draft of this article for review and comment. This is an exploratory article.

**Data Availability Statement:** Not applicable.

**Conflicts of Interest:** The author declares no conflict of interest.

## Notes

1. The Greater New Orleans-Metairie Metropolitan area is inclusive of the Orleans Parish (coterminous with the city of New Orleans) Jefferson Parish, Plaquemines Parish, St. Bernard Parish, St. Charles Parish, St. James Parish, St. John the Baptist Parish and St. Tammany Parish.
2. The Louisiana Department of Children and Families 2015 Human Trafficking of Children for Sexual Purposes, and Commercial Sexual Exploitation Annual Report Pursuant to Act 564 2014 Regular Session does not provide the type of trafficking victimization by Parish.
3. The Louisiana Department of Children and Families 2016, 2017, 2018, 2019, 2020 and 2021 Human Trafficking of Children for Sexual Purposes, and Commercial Sexual Exploitation Annual Report Pursuant to Act 564 2014 Regular Session does not provide the type of trafficking victimization by Parish.

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
