# Peer review of "Conceptualizing Task Force Sustainability"

_societies, doi:10.3390/soc13020050_

Round 1
Reviewer 1 Report
This is a detailed case study and rich explanation of the development of multi-agency approaches to HT in the NO area. The discussion about the sustainability of this team and the reasons for its longevity are interest and well explained. I wondered whether the shape of the nature of HT has changed since the late 2000s when, as explained here, the focus was very much on trafficking for sexual exploitation? As in other jurisdictions, where this has shifted to be much more about forced labour and labour exploitation, it would be interesting to see how and whether this has changed in this area and how this has impacted on the work and composition of the TF. In the discussion section, is there any evidence of whether and how the values of different organisations involved in the TF impacted on the cohesion of the TF? This is well evidenced in literature on other multi-agency work (e.g. domestic abuse) and it would be interesting to know whether there was any evidence of this occurring within these structures. Some more minor comments below - line 46 - 'bringing in prostitutes' rather than 'bringing in prostitution'? line 68 - what period of time does this refer to? The next paragraph skips to 2015 and the previous date is 2008 which seems like a bit of a jump, can you provide some clarity about this here? line 74 - should it be 'which many traffickers preferred' or 'an area which many traffickers preferred'? line 75 - 'as they charged nightly rates' lines 82/83 - are there data about how many prosecutions there have been under this Ordinance that could be added here? line 102 - what time period was this funding for? 2018 until when? Is there any update on the Task Force to date that could be added in here? line 133 - what was the 22 month period? line 135 - are there any examples from the interviews that would evidence the point here? line 217 - the lowering of barriers preventing victims accessing services?Author Response
Thank you so much for your comments. Please see the attached document with responses.

Reviewer 2 Report
The authors demonstrated exceptional knowledge on the subject of human trafficking and the importance of a multi-disciplinary approach. There are a few areas that could be strengthen:
Line 19 referenced a 2014 Louisiana DCFS report, are these the most recent statistics and if not you may want to acknowledge that there are no current stats, also there is no indication if the stats are referring to labor or sex trafficking victims.
Line 173 refers to change is constant. Need to expound on the effects of change on sustainability, in the Discussion or Conclusion. This suggestion is based on Change being a key issue (line 14), the reader can benefit on understanding the theory of change and how it can change the dynamics of a team.
Lines 288-291 this statement could be stronger by explaining how the items listed (leadership, historical knowledge etc.) impact the non-existence of the task force. This would make the conclusion stronger, giving the reader suggestions on how to gaps in current task force can be filled.
Lines 288-291 explain what happens if the task force did not exist, however it omits the contributing factor of lines 184-187 which indicate that "the overall anti-human trafficking work of law enforcement wouldn't change if the Task Force ceased to exist". This contributing factor is important to drive home the commitment of law enforcement to the multidisciplinary approach.
Author Response
Thank you so much for your comments. Please see the attached document with responses.

Reviewer 3 Report
Review of “A Case Study in Task Force Sustainability.”
Societies
This is an underexplored area, and the manuscript is subsequently a potentially important contribution to the research literature.
Abstract- the last sentence needs some restructuring for readability. Maybe “Key focal points related to sustainability explored in this conceptual paper include leadership, funding, collaboration, trust and relationship building, and change as constant.”
Introduction- It is a little confusing when moving from 2014 efforts to 2008. Perhaps some minor reorganization to frame in chronological order would be beneficial.
P. 3—the survey data and qualitative interview data—are the results published elsewhere? If so, include the citations. If not, why not present the results for the special edition as an original article rather than/ in addition to a conceptual paper? I see just a couple of narratives here, leading me to believe an in depth qualitative analysis drawing from the interview data was likely published elsewhere.
Sustainability Process- First sentence, you mention the task foce consulted a number of models—what were they/ briefly describe them and how they informed your work. Or are you referring to the methods used by the task force more broadly? A little clarification is needed.
Self Assessment—can you explain this a bit more? “The members also agreed that collaboration among law enforcement under the Task Force is important, and that there would be a loss in certain law enforcement areas when working cases that benefit from coordinated efforts, such as criminal networks, with a nexus between local, state and Federal jurisdictions.”
The conceptual paper would be further strengthened by providing a succinct implications section, perhaps using the subheadings indicated in the conclusion related to major findings blended with the key points of the discussion. This would provide a scalable easy to read/find guide for readers interested in replicating the key elements found to be beneficial to sustainability. This would mean a slight reorganization and expansion of the discussion/ conclusion.
Author Response

(The authors gave the same response as above.)
